# Nanogold Particles Suppresses 5-Flurouracil-Induced Renal Injury: An Insight into the Modulation of Nrf-2 and Its Downstream Targets, HO-1 and γ-GCS

**DOI:** 10.3390/molecules26247684

**Published:** 2021-12-19

**Authors:** Mohamed El-Sherbiny, Eslam K. Fahmy, Nada H. Eisa, Eman Said, Hany A. Elkattawy, Hasnaa Ali Ebrahim, Nehal M. Elsherbiny, Fatma M. Ghoneim

**Affiliations:** 1Department of Basic Medical Sciences, College of Medicine, Almaarefa University, P.O. Box 71666, Riyadh 11597, Saudi Arabia; msharbini@mcst.edu.sa (M.E.-S.); hmohammed@mcst.edu.sa (H.A.E.); 2Department of Anatomy and Embryology, Faculty of Medicine, Mansoura University, Mansoura 35516, Egypt; 3Medical Physiology Department, College of Medicine, Zagazig University, Zagazig 44519, Egypt; eslam.kamal.fahmy@gmail.com; 4Medical Physiology Department, Faculty of Medicine, Northern Border University, Arar 91431, Saudi Arabia; 5Department of Biochemistry, Faculty of Pharmacy, Mansoura University, Mansoura 35516, Egypt; nadaeisa@hotmail.com; 6Department of Pharmacology and Toxicology, Faculty of Pharmacy, Mansoura University, Mansoura 35516, Egypt; eman-sa3eed@hotmail.com; 7Faculty of Pharmacy, New Mansoura University, New Mansoura 7723730, Egypt; 8Zagazig Obesity Management & Research Unit, College of Medicine, Zagazig University, Zagazig 44519, Egypt; 9Department of Basic Medical Sciences, College of Medicine, Princess Nourah bint Abdulrahman University, P.O. Box 84428, Riyadh 11671, Saudi Arabia; haebrahim@pnu.edu.sa; 10Department of Pharmaceutical Chemistry, Faculty of Pharmacy, University of Tabuk, Tabuk 71491, Saudi Arabia; 11Histology and Cell Biology Department, Faculty of Medicine, Mansoura University, Mansoura 35516, Egypt; fatmaghonaim@gmail.com

**Keywords:** nanogold, 5-flurouracil, Nrf-2

## Abstract

The development of the field of nanotechnology has revolutionized various aspects in the fields of modern sciences. Nano-medicine is one of the primary fields for the application of nanotechnology techniques. The current study sheds light on the reno-protective impacts of gold nano-particles; nanogold (AuNPs) against 5-flurouracil (5-FU)-induced renal toxicity. Indeed, the use of 5-FU has been associated with kidney injury which greatly curbs its therapeutic application. In the current study, 5-FU injection was associated with a significant escalation in the indices of renal injury, i.e., creatinine and urea. Alongside this, histopathological and ultra-histopathological changes confirmed the onset of renal injury. Both gene and/or protein expression of nuclear factor erythroid 2–related factor 2 (Nrf-2) and downstream antioxidant enzymes revealed consistent paralleled anomalies. AuNPs administration induced a significant renal protection on functional, biochemical, and structural levels. Renal expression of the major sensor of the cellular oxidative status Nrf-2 escalated with a paralleled reduction in the renal expression of the other contributor to this axis, known as Kelch-like ECH-associated protein 1 (Keap-1). On the level of the effector downstream targets, heme oxygenase 1 (HO-1) and gamma-glutamylcysteine synthetase (γ-GCS) AuNPs significantly restored their gene and protein expression. Additionally, combination of AuNPs with 5-FU showed better cytotoxic effect on MCF-7 cells compared to monotreatments. Thus, it can be inferred that AuNPs conferred reno-protective impact against 5-FU with an evident modulatory impact on Nrf-2/Keap-1 and its downstream effectors, HO-1 and γ-GCS, suggesting its potential use in 5-FU regimens to improve its therapeutic outcomes and minimize its underlying nephrotoxicity.

## 1. Introduction

5-Fluorouracil (5-FU) is a pyrimidine antimetabolite antineoplastic agent. It possesses remarkable anti-cancer effects against diverse malignancies including breast [1], stomach [2], head and neck [3], colorectal [4], and skin cancers [5]. Intracellularly, 5-FU is converted into three main active metabolites. The first is 5-fluoro-2-deoxyuridine monophosphate which suppresses the synthesis of thymine nucleotide via irreversible inhibition of thymidylate synthase, leading to DNA damage. The second is 5-fluoro-uridine-5′-triphosphate, which is known to incorporate into DNA and RNA replicating strands, interrupting their synthesis. The third metabolite is 5-fluoro-2′-deoxyuridine-5′-triphosphate which also inhibits DNA synthesis via incorporation into DNA and production of single-strand breaks [6].

These events interfere with cellular central dogma with eventual cellular death [7]. However, these actions are not specific and affect both malignant and normal cells, which accounts for 5-FU-induced significant organ toxicities. Myelosuppression [8], gastrointestinal [9], renal [10], and cardiotoxicities [11] are the most famous 5-FU-induced significant organ toxicities. Indeed, 5-FU-induced renal toxicity is a major drawback that curbs its clinical use. 5-FU’s renal toxicity is thought to be mediated by the liver mediated catabolism of 5-FU into dihydrouracil which, in turn, is cleaved into toxic metabolites α-fluoro-β-alanine, ammonia, and urea, thereby leading to nephrotoxicity [12]. Therefore, minimizing side effects of 5-FU has become an urgent need to improve chemotherapy-intended therapeutic outcomes for cancer patients. The mechanism of 5-FU-induced renal toxicity is still not completely verified. However, several investigators proposed 5-FU to trigger excessive free radical production, leading to lipid peroxidation, cell membrane damage, and cellular apoptosis [13,14,15].

Nephron is the functional unit in the kidney. About 1–2.5 million nephrons can be found in adult human kidney [16]. Via its two filtration units (the glomerulus and a hairpin-shaped tubule, the nephron maintains fluid homeostasis, osmoregulation, and waste filtration. The glomeruli consist of glomerular endothelial cells, mesangial cells, podocytes, and glomerular basement membrane and parietal epithelial cells. Glomerular endothelial cells, as well as the basement membrane and podocytes, form a barrier that retains essential components and prevent their loss in the produced urine [17]. Drug-induced renal injury is one of common causes of acute kidney injury and is characterized by high patient morbidity and mortality [18,19,20]. In this context, 5-FU induced glomeruloscelerosis, degeneration of convoluted tubules, and interstitial fibrosis with ultimate renal dysfunction [21].

Currently, an intense focus is directed towards the use of adjuvant therapies to amplify the effectiveness of chemotherapeutic drugs via reducing their induced organ toxicity. Gold is a metal that is well known for its biocompatibility and low cytotoxicity [22]. Gold compounds have been reported to demonstrate anti-inflammatory, antioxidant, and immunomodulatory properties [23]. However, their application is limited due to inactivation by endogenous precipitation and complexation, thereby limiting their demanded functions in human system [24]. Nanotechnology has recently gained increased attention for its wide application in various fields of medicine, in addition to its capability of improving bioavailability and therapeutic efficiency of loaded drugs. For instance, recent studies reported effective anti-viral activities of nanoparticles loaded with antiviral drugs, such as acyclovir fort treatment of herpes simplex virus (HSV) infection [25] and ritonavir for human immunodeficiency virus (HIV-1) [26]. Gold nano-particles (AuNps) are currently renowned for their promising therapeutic efficacies, due to their stable nature, high surface reactivity, biocompatibility, inertness, facile synthesis, and surface plasmon resonance [27,28].

Indeed, AuNPs have been reported to possess antioxidant and anti-inflammatory effects [29]. AuNPs demonstrated antioxidant effects via quenching reactive oxygen species (ROS), such as superoxide anion radical (O_2_^−^) and H_2_O_2_ in a dose-dependent manner [27,30]. Other studies have also revealed anti-hyperglycemic, anti-inflammatory, and antioxidative activities of AUNPs in diabetic animals [24,31], collagen-induced arthritis [32], and bone loss models [33].

In view of current limitations in conventional chemotherapeutic agents, including organ toxicity and multi-drug resistance [34], and the promising reported efficacies of AuNPs, the main goals of the current study are to investigate the potential reno-protective efficacy of AuNPs against 5-FU-induced nephrotoxicity in rats and to explore the possible underlying mechanism. 

## 2. Results

Data collected from both the normal and AuNPs control groups revealed almost no significant difference and, thus, they are collectively referred to and hereafter as normal control.

### 2.1. AuNPs Treatment Attenuated 5-FU-Induced Biochemical and Histopathological Renal Injury

Five days of 5-FU administration at a dose of (50 mg/kg) induced a significant escalation in serum creatinine and urea levels compared to normal control. AuNPs treatment significantly reduced elevated serum creatinine compared to the 5-FU-treated group. By comparing the serum urea between the 5-FU+AuNPs group and the 5-FU group, the *p* value was 0.058, which is almost significant (Figure 1A,B).

The reno-protective impact of AuNPs was further confirmed on a histopathological level. Indeed, histopathological examination confirmed 5-FU-induced severe renal tubular degeneration, necrosis, and glomerular shrinkage. On the other hand, AuNPs treatment significantly improved the above-referred to observed signs of renal structural damage with marked restoration of renal normal cellular structure, as shown in Figure 1C.

Nevertheless, electron microscopic examination of renal cortical tissues of 5-FU control confirmed thickened glomerular basement membrane and widened subpodocytic spaces with irregular nuclear membrane of podocyte. Renal cortical tissues also showed thickened basement membrane of proximal convoluted tubules cells with vacuolated cytoplasm, disarranged basal mitochondria, and irregular nucleus. These 5-FU-induced ultra-structural changes were confirmed to be significantly resolved with AuNPs treatment with marked restoration of the renal ultra-structural characteristics, as shown in Figure 2**A**–**H**.

Representative transmission electron microscopy (TEM) shows images of ultrathin sections of the renal cortex among different groups. Normal control (**A**,**B**) and AUNPs control groups (**C**,**D**) show cells of proximal convoluted tubule rest on the basement membrane (crossed arrow), which have rounded and euchromatic nuclei (N), scattered mitochondria (M), lysosomes (L), and longitudinal oriented basal mitochondria (arrow). The luminal border shows numerous microvilli (MV). The 5-FU group (**E**,**F**) shows the thickened glomerular basement membrane (crossed arrows), widened subpodocytic spaces, and disruption of minor process (asterix) with irregular nuclear membrane of podocyte (arrow heads). Cells of proximal convoluted tubules show thickened basement membrane (crossed arrows), vacuolated cytoplasm (V), disarranged basal mitochondria (arrow), and irregular nucleus N (curved arrow). The 5FU+AUNPs group (**G**,**H**) shows nearly normal glomerular basement membrane (crossed arrows), preserved minor processes (M), major processes (MP), and subpodocytic space (asterix). The cells of proximal convoluted tubules have euchromatic nuclei (N), scattered mitochondria (M), apical microvilli (MV), lysosomes (L), and nearly normal basement membrane (crossed arrow). Uranyl acetate and lead citrate ×2000 scale bar 5 µm and ×1500 scale bar 10 µm. Two different sections (10 fields each) were analyzed (*n* = 6).

### 2.2. AuNPs Treatment Corrected 5-FU-Induced Imbalance of Nrf-2/Keap-1 Axis in Renal Tissues

Nrf-2 and Keap-1 expression usually exhibit a pattern of inverse relationship of expression. 5-FU-induced organ toxicity is accompanied by a marked reduction in Nrf-2 along with an increase in Keap-1. Hence, studies showed that activation of Nrf-2 could be beneficial in the protection against 5-FU-induced organ injury [35,36]. Similarly, data in the present study indicated that 5-FU induced a significant decrease in renal tissue expression of Nrf-2 protein compared to both normal and AuNPs control groups. Upon AuNPs treatment, Nrf-2 protein expression was significantly restored compared to 5-FU control (Figure 3A,B).

On the other hand, and opposite to Nrf-2 expression pattern, 5-FU induced a significant increase in the renal tissue expression of Keap-1 protein compared to both normal and AuNPs controls. Keap-1 tissue upregulation was significantly reversed upon AuNPs treatment (Figure 3C,D).

### 2.3. AuNPs Treatment Restored 5-FU-Induced Suppression of Nrf-2 Downstream Targets, HO-1 and γ-GCS in Renal Tissues

Following the Nrf-2 pathway, differential tissue expression of HO-1 and γ-GCS, which are well-known downstream targets of Nrf-2, was investigated [37]. HO-1 renal mRNA level (Figure 4A), as well as renal tissue protein expression (Figure 4B,C), was significantly decreased in the 5-FU control group compared to the normal control. AuNPs treatment significantly increased renal mRNA and protein expression of HO-1 (Figure 4A–C). Consistently, γ-GCS mRNA level (Figure 4D) and protein expression (Figure 4E,F) was significantly decreased in the renal tissue of the 5-FU group as well. AuNPs treatment significantly increased renal expression of γ-GCS expression on both the gene and protein levels (Figure 4D–F), compared to 5-FU control.

### 2.4. AuNPs Treatment Potentiated 5-FU-Induced Cytotoxic Effect on MCF-7 Cells

The MTT assay was utilized to assess the impact of AuNPs (10, 25, 25 µg/mL) on 5-FU (100 µg/mL)-induced cytotoxicity. As shown in Figure 5, treatment of MCF-7 cells with increasing concentrations of AuNPs produced a marked cytotoxic effect. Further, a combination of AuNPs and 5-FU produced better cytotoxic efficacy compared to mono-treatments.

## 3. Discussion

The revolution that the world has been witnessing lately in the field of nanotechnology has to be considered by the practitioners in the field of medicine. This will introduce novel nanotechnology-based methods to optimize various therapy-intended outcomes, which went on to be used by chemotherapeutic agents for the management of various types of cancers.

Nanotechnology has recently gained increased worldwide attention for its capability to improve bioavailability and therapeutic efficiency of various therapeutic agents with different underlying mechanisms by increasing the uptake of the active agent by target cells; targeting drugs to specific tissues; improving the drug profile, bio-distribution, and pharmacokinetics; and minimizing the associated toxicity [29].

Given their previously referred anti-inflammatory and antioxidant properties, AuNPs represent a future perspective for the treatment of various disorders, including inflammatory diseases and, of particular concern, the number of complications associating the use of various chemotherapeutic agents which mainly stem from their impact on healthy rather than cancerous cells.

5-FU is a widely used chemotherapeutic agent used for the management of various cancers. However, serious toxicity and side-effects occur following its use and it is considered to be a nephrotoxic compound; this greatly curbs its therapeutic application [38]. The results of the current study shed light on the reno-protective effect of AuNPs against 5-FU-induced kidney injury. AuNPs administration to 5-FU-treated rats attenuated the biomarkers of renal injury, i.e., creatinine and urea, by inferring renal functional improvement. Nevertheless, renal specimen revealed significant structural and ultra-structural recovery with AuNPs treatment, inferring structural recovery. Overall, the observed results suggest evident reno-protective impact of AuNPs administration. Interestingly, Vilar et al. (2020) reported that AuNPs attenuated 5-FU-induced experimental oral mucositis, representing another adverse effect commonly encountered in patients receiving chemotherapy [29].

Urea and creatinine are among the most prominent determinants of kidney damage [13,39]. In context, our results showed that IP injection of 5-FU forced a significant increase in the serum biomarkers of renal injury which comes in agreement with observations of Gelen et al. [40]. Thus, it can be inferred that the observed retraction in serum creatinine and urea levels was attributed to a potential reno-protective impact of AuNPs administration.

The mechanism of renal toxicity caused by 5-FU is not completely clear. However, the possible mechanism proposed by several investigators is the overproduction of free radicals, resulting in lipid peroxidation and the subsequent cell membrane damage [13]. The overproduction of ROS and enhanced oxidative status is often paralleled with retraction in the elements of the antioxidant defense batteries. Indeed, 5-FU injection in the current study forced a significant suppression in renal expression of Nrf-2 with a paralleled increase in the expression of Keap,1 which confirms their role in the onset of 5-FU-enhanced oxidative status in kidney.

Oxidative stress is induced by a numerous factors, including drugs and xenobiotics. Enhanced oxidative stress forces the generation of ROS together with various electrophilic molecules which profoundly impact cellular survival and integrity. Amongst the various oxidative stress sensors, Nrf-2 and its effectors are considered to be major protective contributors [41,42].

The Nrf-2 pathway is one of the most important pathways in the cell functioning to protect the various cellular components against oxidative stress [43]. The accumulation of ROS and/or electrophiles enhances oxidative/electrophile stress and cellular membrane damage, degeneration of tissues, and cell death. As a consequence, a battery of defensive gene expression is activated to force the detoxification of ROS and to prevent free radical generation in such a way to enhance cell survival [44]. HO-1 and γ-glutamylcysteine synthetase (γ-GCS) are amongst the defense contributors activated to combat the enhanced oxidative status and enhance ROS elimination. Indeed, Nrf-2 was reported to be involved in the transcriptional activation of γ-GCS and HO-1 antioxidants, proteasomes, and drug transporters [45].

The aforementioned evidence accounts for the observed recovery in the architectural and functional status of the inspected kidney specimen following AuNPs administration. It can be inferred that AuNPs enhanced renal Nrf-2 expression with the subsequent enhancement in the battery of cellular defenses to restore oxidative and antioxidative hemostasis and protect renal tissue against a 5-FU-induced buildup of ROS and enhanced oxidative status.

A cytosolic inhibitor (INrf-2) of Nrf-2, also known as Keap1, was identified. INrf-2, or Keap1, retains Nrf-2 in the cytoplasm [46]. When a cell is exposed to oxidative stress, Nrf-2 dissociates from the INrf-2 complex, stabilizes, and translocates into the nucleus, leading to activation of antioxidant response element (ARE)-mediated gene expression [45]. Another theory is that Nrf-2, in response to oxidative stress, escapes Keap1 degradation, stabilizes, and translocates to the nucleus with subsequent activation of ARE [47]. Following either theory, the net result enhances the antioxidant response with enhanced elimination of ROS and other electrophilic molecules, contributing to the preservation of cell membrane integrity and preservation of cellular and organs’ functions. 

AuNPs have been reported to be involved in regulating the Keap1 pathway, and adjusting the cytoprotective response to endogenous and exogenous stress caused by ROS [48]. Under baseline conditions, the Keap1 repressor protein binds to the inactive NRF-2 factor present in the cytoplasm. In the presence of ROS, Keap1 releases Nrf-2 factor which translocates to the nucleus, and interacts with the ARE, inducing increased expression of various effectors including HO-1, NAD(P)H quinone oxidoreductase 1 (NQO1), glutathione (GPx), and superoxide dismutase (SOD), resulting in an antioxidant response [49,50].

Given the inverse relationship between renal Nrf2 and Keap1 tissue expression, the previously mentioned evidence reveals the observed increased expression of Keap1 in the renal specimen of 5-FU-treated rats which account for its inhibitory effect on Nrf-2 expression. Thus, it can be inferred that AuNPs administration via decreasing tissue expression of the Keap1 enhanced the expression of Nrf-2 which in turn activated the antioxidant defense elements and restored the renal oxidative status with mitigation of renal structural and ultra-structural injuries and preservation of renal architecture and functions.

In this regard, 5-FU injection, in the current study, depleted renal HO-1 and γ-GCS gene and protein expression. On the other hand, AuNPs forced a significant increase in tissue and gene expression of both of HO-1 and γ-GCS, i.e., the downstream target effectors of Nrf-2. Indeed, AuNPs induced HO-1 protein and mRNA expression in a concentration- and time-dependent manner in human vascular endothelial cells [48]. These observations reported by Lai et al. gives credence to the observed results from the present study as they reported that, in response to the AuNP treatment, the cytosolic Nrf-2 translocated to the nucleus, and the translocated Nrf-2 bound to the antioxidant-response element located in the E2 enhancer region of the HO-1 gene promoter and acted as a transcription factor. 

HO is the rate-limiting enzyme in the catabolism of heme [51]. It has two isoenzymes: the original enzyme; HO-1, and the second isoenzyme; HO-2. The anti-inflammatory properties of HO-1 have been attributed to its down-regulatory impact on TNF-α-induced expression of various adhesion molecules [52]. Furthermore, AuNPs have been reported to enhance HO-1 expression in rat aortic vascular smooth muscle cells by inducing Nrf-2 expression, phosphorylation, and translocation into nucleus [53]. 

With regard to γ-GCS, reduced glutathione (GSH) is a ubiquitous tripeptide thiol, vital intracellular, and extracellular protective antioxidant, playing a crucial role in the detoxification of various xenobiotics [54]. GSH is synthesized in the cell by γ-GCS and glutathione synthetase. The γ-GCS-catalyzed formation of γ-glutamylcysteine is the first and rate-limiting step in de novo GSH synthesis and is feedback-inhibited by GSH, i.e., the central mechanism for the regulation of cellular GSH concentrations [55].

In the current study, 5-FU administration was associated with decreased protein and gene expression of γ-GCS inferring reduced cellular capacity to provide and replenish intracellular GSH stores which can be presumed to induce a failure in the cellular response to combat 5-FU-induced enhanced oxidative injury with evident prevalence of kidney injury. On the other hand, AuNPs managed to successfully restore both protein and gene expression of γ-GCS with subsequent enhancement in the antioxidative status and preservation of renal integrity and functions. Such an observation further strengthens and confirms the initial hypothesis about the role of the antioxidant properties of AuNPs on suppressing 5-FU-induced enhanced oxidative status and kidney injury with subsequent evident reno-protective impact. To the best of our knowledge, this is the first study reporting the impact of AuNPs on γ-GCS.

## 4. Materials and Methods

### 4.1. Drugs and Chemicals

5-FU was purchased as 5-flurouracil 250 mg 5 mL/ampoule (Dawaya Co., Cairo, Egypt). AuNPs were purchased from PlasmaChem GmbH, Germany (Catalogue No: PL-Au-S20, 0.05 mg/mL aqueous solution). The average particle size was 20 nm. According to manufacturer, the nanoparticle colloidal solution was prepared by citrate sol–gel reduction, and the particle shape was spherical with a negative surface charge through the citrate stabilization. AuNPs were administered to the rats by intraperitoneal (IP) injection.

### 4.2. Animals and Experimental Design

Adult Sprague Dawley rats (160–180 g weight and 8 weeks age) were obtained from the breeding unit of Holding Company for Biological Products and Vaccines, “VACSERA”, and were housed in clean plastic cages under standard laboratory conditions (temperature of 25 ± 2 °C, relative humidity of 50–60%, and an alternating 12-h dark–light cycle). All experimental procedures were approved by the Research Ethics Committee, Faculty of Medicine, Mansoura University, Egypt.

Following one week of adaptation, animals were randomly divided into 4 groups (with 6 rats/group) as follows. In **Group I (normal control),** rats received vehicle (normal saline) for 28 days; in **Group II (AuNPs control group),** rats received 70 μg/kg of AuNps IP for 28 days; **Group III (5-FU control group),** rats received vehicle for 23 days, followed by 50 mg/kg of 5-FU IP for five days (from day 24 to day 28); and in **Group IV (5-FU+AuNPs),** rats received 70 μg/kg of AuNps IP for 23 days and 50 mg/kg of 5-FU IP for 5 days (from day 24 to day 28). Doses for 5-FU and AuNPs were selected based on preliminary studies and previous reports [56,57].

At the end of the experimental period, rats were sacrificed with thiopental overdose (40 mg/kg, IP) and blood samples were withdrawn from the retro-orbital sinuses using capillary tubes. Serum samples were separated by centrifugation at 3000 rpm for 5 min. The left and right kidneys were dissected out, split lengthwise, and washed with ice-cold saline. Half of the left kidney specimen was preserved in 10% buffered formalin for histopathological assessments, and the other half part was fixed in glutaraldehyde buffer (2.5%) for electron microscopy examination. The right kidney was flash-frozen in liquid nitrogen and stored at −80 °C for further real-time polymerase chain reaction (RT-PCR) analysis.

### 4.3. Assessment of Serum Markers of Renal Injury; Creatinine and Urea

Serum concentrations of creatinine (#CR 1251) and urea (#UR 2110) were measured by the spectrophotometric method, according to manufacturers’ procedures using commercially available kits supplied by Biodiagnostic (Giza, Egypt).

### 4.4. Real Time Polymerase Chain Reaction (RT-PCR)

Total RNA was extracted from kidney tissues using the Direct-zol RNA Miniprep Plus kit (Cat# R2072, Zymo Research Corp., Irvine, CA, USA), and then the quantity and quality of RNA were assessed with the Beckman dual spectrophotometer (USA). The SuperScript IV One-Step RT-PCR kit (Cat# 12594100, Thermo Fisher Scientific, Waltham, MA, USA) was utilized for reverse transcription of extracted RNA, followed by PCR, in one step. The reaction mixture included an RNA sample, SuperScript™ IV RT Mix, Platinum™ SuperFi™ RT-PCR Master Mix, and forward and backward primers of HO-1 and γ-GCS genes (Table 1). The prepared reaction mix samples were applied in RT-PCR (Step One Applied Biosystem, Foster City, CA, USA). After the RT-PCR run, the data were expressed in the cycle threshold (Ct). The relative quantification of each target gene was quantified and normalized to the housekeeping gene, known as GAPDH (Table 1), by calculation of 2^−∆∆Ct^.

### 4.5. Histological Assessment

For evaluation of the renal structure, formalin-fixed specimens were washed, dehydrated, cleared, and embedded in paraffin. The blocks were further used to prepare 5-µm sections, which were mounted on glass slides and then stained with hematoxylin and eosin (H&E). Slides were analyzed at 100× and 400× magnification using Olympus^®^ light microscope, and images were captured using Olympus^®^ digital camera (City, Tokyo, Japan).

### 4.6. Immunohistochemical Analysis of Renal Expression of Nuclear Factor Erythroid 2–Related Factor 2 (Nrf-2), Gamma-Glutamylcysteine Synthetase (γ-GCSc), Kelch-like ECH-Associated Protein 1 (Keap1) and Heme Oxygenase-1 (HO-1)

Sections with 5-μm thickness were mounted on slides and then deparaffinized using xylene and ethanol solutions of different grades. The slides were then treated with citrate buffer (pH = 6) for antigen retrieval, blocked using 1% bovine serum albumin. Thereafter, the slides were incubated at 4 °C overnight with primary antibodies for Nrf-2 (ab89443), γ-GCSc (sc-166356), Keap1 (sc-514914), and HO-1 (sc-390991). Following washing, a species-matched secondary antibody was added for 1 h at room temperature, and the reaction was visualized using an ABC system. Sections were counterstained using Mayer’s hematoxylin. Dibutylphthalate polystyrene xylene (DPX) was used as mounting medium. Images were captured at 400× magnification using Olympus^®^ digital camera (Tokyo, Japan) fixed on an Olympus light microscope. Area percentage of immunoreactivity was calculated using ImageJ software (NIH, Bethesda, MD, USA). 

### 4.7. Electron Microscopy Investigation

Glutaraldehyde buffer fixed specimens were further incubated for 1 h in osmium tetroxide 1.0%. The specimens were then washed, dehydrated using alcohol solutions of different grades, and then embedded in a 1:1 ratio of propylene and epoxy resin, followed by epoxy resin. The blocks were then used to cut ultra-thin sections, which were further stained with uranyl acetate and lead citrate. The photographs were then taken by transmission electron microscope (JEOL 2100, Tokyo, Japan).

### 4.8. MTT Assay

MCF-7 human breast carcinoma cells were seeded into 96-well culture plate at densities 1 × 10^3^ cells per well and incubated for 48 h in 5% CO_2_ incubator. Thereafter, the seeded cells were treated with AuNPs (10, 25, 50 µg/mL) and 5-FU (100 µg/mL) for 24 h. The MTT assay was performed, as previously described [58]. Briefly, 10 µL of MTT solution (12-mm MTT stock solution) was diluted by serum-free media and added to each well. The plate was then incubated for 4 h at 37 °C. Following incubation, 100 µL of detergent solution (SDS-HCl) was added and the plate was incubated for 18 h at 37 °C. The color absorbance was measured at 590 nm using an ELISA plate reader. The amount of absorbance represents cell viability. 

### 4.9. Statistical Analysis

Results are presented as mean ± SD. Statistical significance among different experimental groups was evaluated using a one-way ANOVA test followed by Tukey’s post-hoc test. GraphPad prism software (San Diego, CA, USA) was used for statistical analysis and data presentation. *p* values less than 0.05 were considered statistically significant.

## 5. Conclusions

In conclusion, AuNPs protected against 5-FU-induced kidney injury through Nrf-2/Keap-1 signaling. The underlying protective mechanisms involved inhibition of Keap-1, augmentation of Nrf-2 signaling, and enhancement of downstream effectors HO-1 and γ-GCS with subsequent restoration of the cellular oxidative status and ultimate renal functional and biochemical protection. AuNPs are suggested to be incorporated in 5-FU-containing therapeutic regimens to enhance 5-FU-associated therapeutic outcomes and minimize its underlying nephrotoxicity.

## Figures and Tables

**Figure 1 molecules-26-07684-f001:**
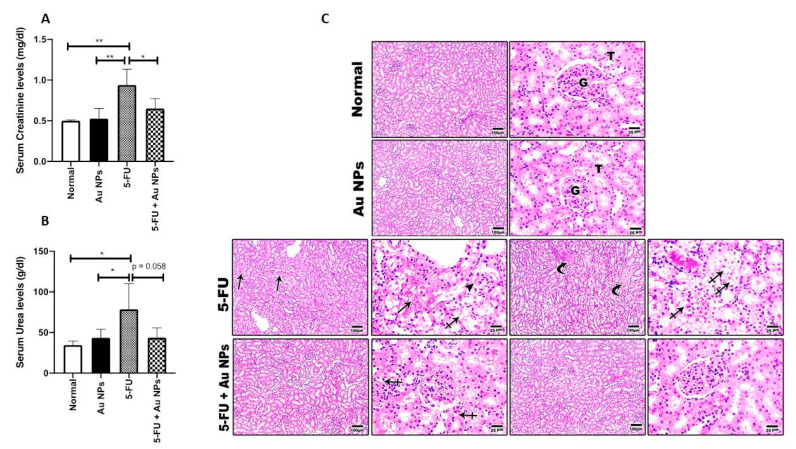
Effect of AuNPs treatment on 5-FU-induced renal injury. AuNPs significantly reduced 5-FU-elevated serum creatinine (**A**) and urea (**B**) levels (*n* = 4–6). * *p* < 0.05, ** *p* < 0.01 indicate statistical significance compared with the 5-FU group. (**C**) Representative microscopic images of H&E stained renal sections, showing normal glomeruli (G) and tubules (T) with minimal interstitial tissue in control normal and AuNPs control groups, diffuse severe tubular degeneration (crossed arrows), multifocal tubular necrosis (arrows), glomerular shrinkage (arrowhead), interstitial fibrosis (curved arrows) in the 5-FU group. Mild tubular degeneration (crossed arrows) and great improvement of histology of glomeruli (G) and tubules (T) with minimal interstitial tissue was also shown in the 5-FU+AuNPs group. Low magnification (100×) and high magnification (400×) were used for acquiring images with indicated scale bar. Two different sections (10 fields each) were analyzed (*n* = 6).

**Figure 2 molecules-26-07684-f002:**
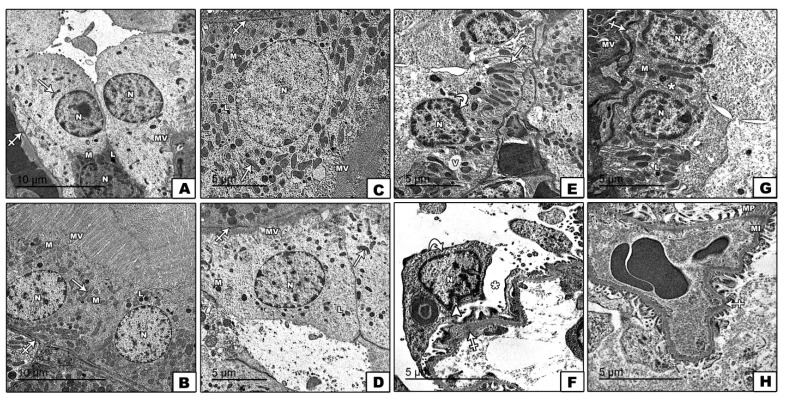
Effect of AuNPs treatment on renal cortical ultrastructural changes induced by 5-FU (**A–H**).

**Figure 3 molecules-26-07684-f003:**
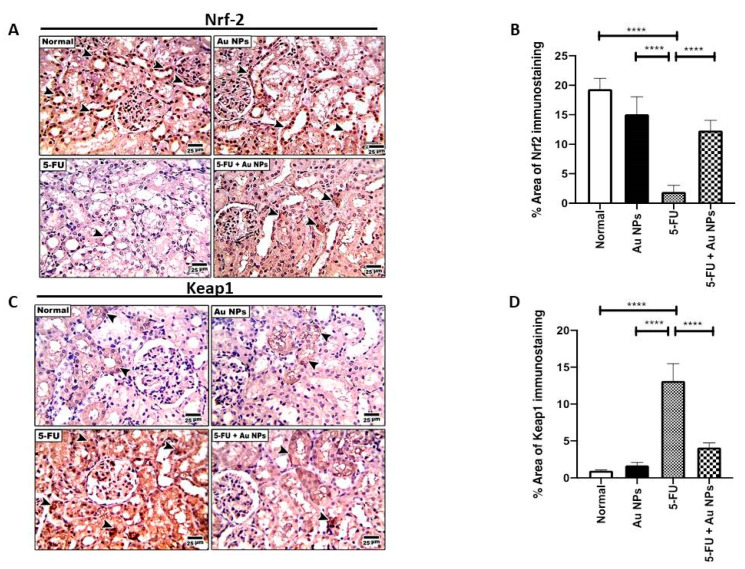
Effect of AuNPs treatment on differential expression of Nrf-2/Keap-1 axis. (**A**) Representative microscopic images of immunohistochemical staining show the reduced expression of the Nrf-2 in the 5-FU group compared with normal and AuNPs control groups (Arrow heads). AuNPs treatment showed significant restoration of Nrf-2 expression in the 5-FU+ AuNPs group compared with the 5-FU group (Arrow heads). High magnification (400×) was used with a scale bar (25 µm). (**B**) Percentage area of Nrf-2 immunostaining (*n* = 6). **** *p* < 0.0001 indicates statistical significance compared with the 5-FU group. (**C**) Representative microscopic images of immunohistochemical staining showing increased expression of Keap-1 in the 5-FU group compared with normal and AuNPs control groups (Arrow heads). AuNPs treatment showed significant reduction in Keap-1 expression in the 5-FU+AuNPs group compared with the 5-FU group (Arrow heads). High magnification (400×) was used with scale bar indicated (25 µm). (**D**) Percentage area of Keap-1 immunostaining (*n* = 6). **** *p* < 0.0001 indicate statistical significance compared with the 5-FU group.

**Figure 4 molecules-26-07684-f004:**
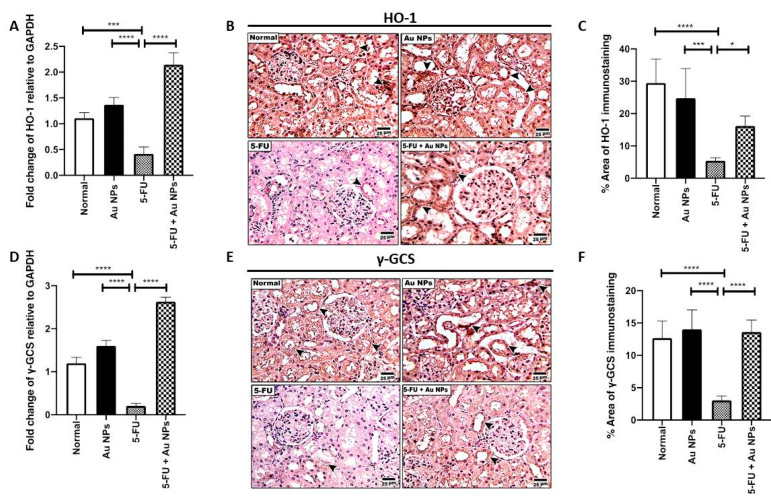
Effect of AuNPs treatment on differential expression of HO-1 and γ-GCS downstream molecules of Nrf-2. (**A**) Fold change in HO-1 mRNA levels relative to GAPDH, as determined by real time PCR (*n* = 4). *** *p* < 0.001, **** *p* < 0.0001 indicate statistical significance compared with the 5-FU group. (**B**) Representative microscopic images of immunohistochemical staining showing reduced expression of HO-1 in the 5-FU group compared with normal and AuNPs control groups (Arrow heads). AuNPs treatment showed significant restoration of HO-1 expression in the 5-FU+AuNPs group compared with the 5-FU group (Arrow heads). High magnification (400X) was used with scale bar indicated (25 µm). (**C**) Percentage area of HO-1 immunostaining (*n* = 6). * *p* < 0.05, *** *p* < 0.001, **** *p* < 0.0001 indicate statistical significance compared with the 5-FU group. (**D**) Fold change in γ-GCS mRNA levels relative to GAPDH as determined by real time PCR (*n* = 4). **** *p* < 0.0001 indicate statistical significance compared with the 5-FU group. (**E**) Representative microscopic images of immunohistochemical staining showing reduced expression of γ-GCS in the 5-FU group compared with the normal and AuNPs control groups (Arrow heads). AuNPs treatment showed significant restoration of γ-GCS expression in the 5-FU+AuNPs group compared with the 5-FU group (Arrow heads). High magnification (400×) was used with a scale bar (25 µm). (**F**) Percentage area of γ-GCS immunostaining (*n* = 6). **** *p* < 0.0001 indicate statistical significance compared with the 5-FU group.

**Figure 5 molecules-26-07684-f005:**
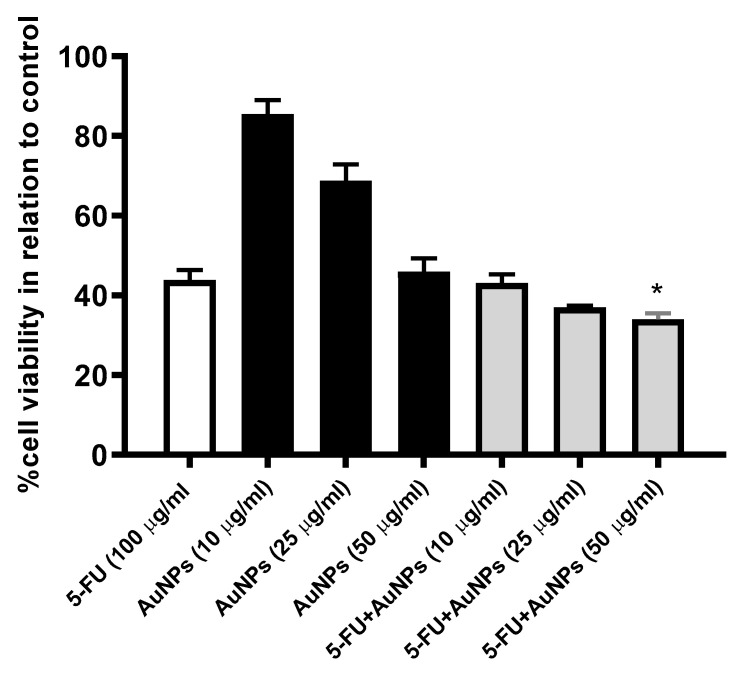
Effect of 24 treatment with 5-FU (100 µg/mL), AuNPs (10, 25, 25 µg/mL), and their combination on MCF-7 cells viability. Results are presented as mean ± SE. * *p* < 0.05 compared to 5-FU (100 µg/mL)-treated cells.

**Table 1 molecules-26-07684-t001:** Primers’ sequences of all studied genes.

Gene Symbol	Forward	Reverse	Gene Bank
HO-1	5′-GAGCGCCCACAGCTCGACAG-3′	5′-GTGGGCCACCAGCAGCTCAG-3′	XM_032887931.1
γ-GCS	5′-AGACACGGCATCCTCCAGTT-3′	5′-CTGACACGTAGCCTCGGTAA-3′	NM_012815.2
GAPDH	5′-ATGGTGAAGGTCGGTGTGAACG-3′	5′-TGGTGAAGACGCCAGTAGACTC-3′	XM_017592435.10

## Data Availability

Data supporting reported results may be supplied upon request to the authors.

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
