# Peer review of "Nanogold Particles Suppresses 5-Flurouracil-Induced Renal Injury: An Insight into the Modulation of Nrf-2 and Its Downstream Targets, HO-1 and γ-GCS"

_molecules, 2021, doi:10.3390/molecules26247684_

Round 1

Reviewer 1 Report

The authors of  manuscript "NanoGold Particles Suppresses 5-Flurouracil-Induced Renal Injury: An Insight into the Modulation of Nrf-2 and its downstream targets, HO-1 and γ-GCS", claim to the  use of NanoGold Particles as previous treatment at 5-FU, decreased kidney damage in a in vivo model. The authors present relevant evidences in renal tissues that confirm the protective effects of the AuNPs. But, I have some questions and suggestions that need to be clarified.

1.Some areas could benefit from being revised for clarity and grammar, most notably:

-The abstract, line 32-33: "Both gene and protein expression of various effector proteins and genes revealed consistent paralleled anomalies".

This tex is confusing and Its unclear which proteins and genes were evaluated.

-Line 34: The authors claim that use of  AuNPs induced a significant renal recovery, however, in this manuscript does not show data in which it has been used 5-FU as first treatment and then the AuNPs, so to confirm the recovery in damage tissues. In this context, the data suggest a protective but no recovery effect.

-Line 44-61: the paragraph is large and lacks bibliography, is necessary citations to give credence to details such as effects of 5-FU in different cancer types 

-Line 94-95: "AuNPs reduced 5-FU-induced serum creatinine and urea elevation" It is unclear.

-Line 101: restoration of renal architectural??

-Figure 1 b)- p=0.058? significant or not significant? not indicated in the figure legend this p value.

-Line 112: Cisplatin group?? This chemotherapeutic is not indicated in the methodology.

-Line 131: Empty parentheses in figure legend.

-2.2. AuNPs treatment corrected 5-FU-induced imbalance of Nrf-2/Keap-1 axis in renal tissues:

It would be appropriate to include a brief description of the relevance of proteins evaluated in this section and connection with AuNPs effects.

-Figure 3C. Labels are confusing, Flu+Au. Flu labels are missing in figure legends.

-Line 164: Restored RNA? maintained or increased , the RNA data with 5-FU as first treatment its not shown.

-line 168: restored.

The conclusions could be improved and include the findings importance and  relevance of AuNPs. 

I suggest a short assay with  AuNPs+5-FU in cancerous cell lines, to confirm that chemotherapeutic effect of 5-FU in cancerous cells is not modified, so to be able to conclude that   AuNPs could be a potential adjuvant in the 5-FU therapy that enhances 5-FU associated outcomes and minimizing its underlying nephrotoxicity.

Author Response

Reviewer 1

The authors of manuscript "NanoGold Particles Suppresses 5-Flurouracil-Induced Renal Injury: An Insight into the Modulation of Nrf-2 and its downstream targets, HO-1 and γ-GCS", claim to the  use of NanoGold Particles as previous treatment at 5-FU, decreased kidney damage in a in vivo model. The authors present relevant evidences in renal tissues that confirm the protective effects of the AuNPs. But, I have some questions and suggestions that need to be clarified.

1.Some areas could benefit from being revised for clarity and grammar, most notably:

-The abstract, line 32-33: "Both gene and protein expression of various effector proteins and genes revealed consistent paralleled anomalies". This tex is confusing and Its unclear which proteins and genes were evaluated.

Thank you for your comment. The sentence was rephrased into “Both gene and/or protein expression of Nrf-2 and downstream antioxidant enzymes revealed consistent paralleled anomalies”.

-Line 34: The authors claim that use of  AuNPs induced a significant renal recovery, however, in this manuscript does not show data in which it has been used 5-FU as first treatment and then the AuNPs, so to confirm the recovery in damage tissues. In this context, the data suggest a protective but no recovery effect.

Thank you for your comment. Recovery was replaced by protection.

-Line 44-61: the paragraph is large and lacks bibliography, is necessary citations to give credence to details such as effects of 5-FU in different cancer types

Thank you for your suggestion. More bibliography was added.

-Line 94-95: "AuNPs reduced 5-FU-induced serum creatinine and urea elevation" It is unclear.

Thank you for your comment. The sentence was rephrased and clarified

-Line 101: restoration of renal architectural??

The sentence was changed to restoration of renal normal cellular structure.

-Figure 1 b)- p=0.058? significant or not significant? not indicated in the figure legend this p value.

It is almost significant based on p value ˂0.05. Regarding serum urea, on comparison of 5-FU+AuNPs group with 5-FU group, p value was 0.058 which is almost significant. This sentence was added to results.

-Line 112: Cisplatin group?? This chemotherapeutic is not indicated in the methodology. We apologize for this mistake, it was corrected to 5-FU

-Line 131: Empty parentheses in figure legend.

Thank you very much for your careful revision. The empty parentheses should contain crossed arrows

-2.2. AuNPs treatment corrected 5-FU-induced imbalance of Nrf-2/Keap-1 axis in renal tissues:

It would be appropriate to include a brief description of the relevance of proteins evaluated in this section and connection with AuNPs effects.

Thank you for your suggestion. A paragraph was added

5-FU induced organ toxicity is accompanied by marked reduction of Nrf-2 along with increase in Keap-1. Hence, studies showed that activation of Nrf-2 could be beneficial in protection against 5-FU induced organ injury (26), (27).

  1. Zeng D, Wang Y, Chen Y, Li D, Li G, Xiao H, et al. Angelica Polysaccharide Antagonizes 5-FU-Induced Oxidative Stress Injury to Reduce Apoptosis in the Liver Through Nrf2 Pathway. Front Oncol. 2021;11:720620.
  2. Wang C, Huo X, Gao L, Sun G, Li C. Hepatoprotective Effect of Carboxymethyl Pachyman in Fluorouracil-Treated CT26-Bearing Mice. Molecules. 2017 May 6;22(5):E756.

-Figure 3C. Labels are confusing, Flu+Au. Flu labels are missing in figure legends.

Thank you for your suggestion. Labels were modified

-Line 164: Restored RNA? maintained or increased , the RNA data with 5-FU as first treatment its not shown.

Thank you for your comment.  Restored was changed to increase. mRNA level in 5-FU was clarified

-line 168: restored.

Thank you for your comment.  Restored was changed to increase.

The conclusions could be improved and include the findings importance and  relevance of AuNPs. 

Thank you for your suggestion. The conclusion was improved

I suggest a short assay with  AuNPs+5-FU in cancerous cell lines, to confirm that chemotherapeutic effect of 5-FU in cancerous cells is not modified, so to be able to conclude that   AuNPs could be a potential adjuvant in the 5-FU therapy that enhances 5-FU associated outcomes and minimizing its underlying nephrotoxicity.

Thank you for your suggestion. MTT assay was utilized to assess the impact of AuNPs (10, 25, 25 µg/ml) on 5-FU (100 µg/ml)-induced cytotoxicity. As shown in Figure 5, treatment of MCF-7 cells with increasing concentrations of AuNPs produced marked cytotoxic effect. Further, combination of AuNPs and 5-FU has produced better cytotoxic efficacy compared to mono-treatments. 

Reviewer 2 Report

  1. Please define the abbreviations the first time they appear in such as Keap-1 line 36, HO-1 and γ-GCS, etc
  2. Please be consistent sometimes nano-particles others nanoparticles; anti-oxidant and antioxidant, etc
  3. Line 27 “impact” should be impacts
  4. Please recheck the reference formatting for example multiple references should appear within one bracket, line 65, line 78, etc
  5. Please revise the subscripts and superscripts across the manuscript
  6. Line 93 what was the basis of the 50 mg/kg dose selection?
  7. Figure 1 please change the scale bar to black for clarity. The arrows are not clear as well
  8. Figure 2 the labeling within the panels on the tissues are not clear please improve the figure quality and labeling for clarity
  9. Please use the correct symbol unit for example micrometer not (uM)
  10. Please include the IRB for the animal studies and refer to it in the methods section
  11. What was the concentration of AuNPs used for the animal studies
  12. Did the author quantify the protein expression via western blot analysis?
  13. Authors should include a short paragraph at the end of the introduction about current limitations in conventional cancer therapies
  14. Authors are encouraged to include a paragraph on renal anatomy and selective permeability
  15. Could the authors include more information from the supplier about the charge and the shape of NPs as they are important factors that influence the distribution of the particles?
  16. A graphical abstract is missing
  17. Methods section please state the concentration, measurement angle, laser type, detector, and wavelength for ZP include the attenuation, temperature, and instrument settings while recording the measurements, and please mention the number of measurements taken for each sample to find the average zeta potential values
  18. The introduction does not capture the massive number of papers in the literature on the SNL particles loaded with various antiviral drugs. Furthermore, it should clearly state the added value from this approach
  19. The manuscript would benefit greatly from English and grammar check. Many typos across the manuscript
  20. Please include the following related references
    1. https://doi.org/10.3390/pharmaceutics12040304
    2. 1007/s13346-019-00675-6
    3. https://doi.org/10.1021/acs.nanolett.1c01840

Author Response

Please define the abbreviations the first time they appear in such as Keap-1 line 36, HO-1 and γ-GCS, etc

Thank you for your comment. The abbreviations were defined

Please be consistent sometimes nano-particles others nanoparticles; anti-oxidant and antioxidant, etc

Thank you for your careful revision. Consistency was considered through the manuscript

Line 27 “impact” should be impacts

Impact was changed to impacts

Please recheck the reference formatting for example multiple references should appear within one bracket, line 65, line 78, etc

Thank you, the references were changed to be in one bracket

 Please revise the subscripts and superscripts across the manuscript

Thank you for your comment. The subscripts and superscripts were revised across the manuscript

 Line 93 what was the basis of the 50 mg/kg dose selection?

Doses for 5-FU and AuNPs were selected based on preliminary studies and previous reports.

Medeiros A da C, Azevedo ÍM, Lima ML, Araújo Filho I, Moreira MD. Effects of simvastatin on 5-fluorouracil-induced gastrointestinal mucositis in rats. Rev Col Bras Cir. 2018 Oct 18;45(5):e1968.

Ferreira GK, Cardoso E, Vuolo FS, Michels M, Zanoni ET, Carvalho-Silva M, et al. Gold nanoparticles alter parameters of oxidative stress and energy metabolism in organs of adult rats. Biochem Cell Biol. 2015 Dec;93(6):548–57.

 Figure 1 please change the scale bar to black for clarity. The arrows are not clear as well

Thank you. Changes have been made as suggested

Figure 2 the labeling within the panels on the tissues are not clear please improve the figure quality and labeling for clarity

Thank you for your comment. The figure quality was improved.

 Please use the correct symbol unit for example micrometer not (uM)

The symbols were corrected.

Please include the IRB for the animal studies and refer to it in the methods section

Thank you for your comment. The IRB was included.

What was the concentration of AuNPs used for the animal studies

 The dose was 70 μg/kg and the concentration was 0.05 mg/ml aqueous solution (added to materials and methods section).

Did the author quantify the protein expression via western blot analysis?

We assessed gene expression by RT-PCR and protein expression with immunohistochemistry and calculation of % area of immunostaining for quantification

Authors should include a short paragraph at the end of the introduction about current limitations in conventional cancer therapies

Thank you for your suggestion. A short paragraph was included

Authors are encouraged to include a paragraph on renal anatomy and selective permeability

Thank you for your suggestion. A paragraph was included on renal anatomy, permeability and effect of various drugs on renal function.

Could the authors include more information from the supplier about the charge and the shape of NPs as they are important factors that influence the distribution of the particles?

Particle shape is spherical with a negative surface charge through the citrate stabilization.

A graphical abstract is missing

Thank you for your suggestion. A graphical abstract was added.

Methods section please state the concentration, measurement angle, laser type, detector, and wavelength for ZP include the attenuation, temperature, and instrument settings while recording the measurements, and please mention the number of measurements taken for each sample to find the average zeta potential values

As per supplier information, unfortunately, the supplier can’t measure zeta potentials ourselves, lacking the device, so the supplier don’t have own data on that. The supplier only know from other customers results, that ZP is approx. -30 mV. The TDS with the material information available at present is attached as supplementary material.

 The introduction does not capture the massive number of papers in the literature on the SNL particles loaded with various antiviral drugs. Furthermore, it should clearly state the added value from this approach

Thank you for your suggestion. Information about SNL particles loaded with various antiviral drugs was added to introduction

 The manuscript would benefit greatly from English and grammar check. Many typos across the manuscript

Thank you for your careful revision. The manuscript was revised and carefully checked for English and grammar mistakes

 Please include the following related references

https://doi.org/10.3390/pharmaceutics12040304

1007/s13346-019-00675-6

https://doi.org/10.1021/acs.nanolett.1c01840

Thank you for your suggestion. References were added.

Round 2

Reviewer 2 Report

No more comments